# Quantitative and Qualitative Identification of Bioactive Compounds in Edible Flowers of Black and Bristly Locust and Their Antioxidant Activity

**DOI:** 10.3390/biom10121603

**Published:** 2020-11-26

**Authors:** Ewelina Hallmann

**Affiliations:** Department of Functional and Organic Food, Institute of Human Nutrition Sciences, Warsaw University of Life Sciences, Nowoursynowska 159c, 02-776 Warsaw, Poland; ewelina_hallmann@sggw.edu.pl; Tel.: +48-22-59-370-36

**Keywords:** black locust, bristly locust, flower, polyphenols, antioxidant activity

## Abstract

Black and bristly locust flowers are an excellent source of polyphenols, including flavonols, phenolic acids, and anthocyanins. In the present literature, there is a lack of studies showing the quantity and quality of phenolic compounds from different locust flowers. There are a few studies on the status of polyphenols in black locust flowers and their products but not bristly locusts. The aims of this study were to analyze and compare the concentrations of bioactive compounds from *Robinia pseudoacacia* and *Robinia hispida* flowers over two years. These two species of plants from six independent locations (parks and green areas) located in Warsaw were assessed in this study. The dry matter and polyphenol contents of the flowers were determined. Black locust flower samples contained significantly more myricetin and luteolin. Only bristly locust flowers contained anthocyanins. Five individual anthocyanins were identified in the pink-colored bristly locust flowers. Pelargonidin-3-*O*-glucoside and cyanidin-3-*O*-glucoside were the predominant forms in the pool of total anthocyanins.

## 1. Introduction

*Robinia hispida*, known better as the bristly locust, and *Robinia pseudoacacia* L., commonly known as the black locust, belong to the Fabaceae family. In Poland, both locus trees are planted mostly as ornamental trees that have white and pink flowers. In time of flowering (May and June), black locust flowers are one of the most important sources of honey production [1,2,3]. Black locust wood is commonly used for wine barrel production. Similar to oak wood, it gives wine many bioactive compounds that enrich the odor and taste of the wine [4,5]. Similar to fruits and vegetables, the flowers of many species are also considered as edible. In the consumer diet, they could be a fine source of different bioactive compounds. Multi-colored flowers have a large concentration of bioactive compounds. Depending on the color of the flowers, different phytochemical profiles were observed, with yellow and orange flowers displaying higher concentrations of carotenoids and flavonoids, while purple, violet, and blue colored flowers displaying higher quantities of anthocyanins [6,7,8,9,10,11]. Of course, the first purpose of flower coloring is to make the flowers more attractive to pollinating insects [12]. In black locust flowers, many bioactive and volatile compounds have been found [13]. In the huge group of polyphenols compounds, quercetin and their glucodsides, epigallocatechin and ferulic acid, are the most common [14,15]. On the other hand, flowers can be a source of polyphenol compounds in the human diet. There are many species with edible flowers and high bioactive compound concentrations [16,17]. Bioactive compounds from edible flowers have potential health benefits. Higher polyphenol consumption can prevent different kinds of cancer, reduce cardiovascular diseases, and decrease blood cholesterol levels [18,19,20]. In traditional cuisine, black locust and bristly locust flowers can be consumed as concentrates, syrups, flower flour, and fried in pancakes, similar to elderberry flowers [21]. In the literature, there is a lack of information on the qualitative and quantitative composition of polyphenolic compounds from the edible flowers of black and especially bristly locusts in addition to the variability in the content of these compounds between individual years. Only a few experiments with total polyphenols have been done with flowers of black locust, not bristly locust flowers. On the other hand, there is a limited amount of information about the profile of phenolic compounds in both species. Therefore, the current studies were conducted to show the composition of biologically active compounds of edible *Robinia* flowers.

## 2. Materials and Methods

### 2.1. Materials and Samples

#### 2.1.1. Chemicals

ABTS (2,2′-Azino-bis(3-ethylbenzothiazoline-6-sulfonic acid) diammonium salt, CAS no.: 30931-67-0) was obtained from Sigma-Aldrich (Warsaw, Poland), acetonitrile (CAS no.: 75-05-8) from Alchem (Poland), deionized water (made in Dejonizator BASIC 5), and hydrochloric acid (35% pure for analysis, CAS no.: 7647-01-0) were obtained from Chempur (Warsaw, Poland); methanol (CAS no.: 67-56-1) was obtained from Sigma-Aldrich (Poland); phenolic standards (HPLC grade) were obtained from Sigma-Aldrich (Poland); phenolic acids: gallic (CAS no.: 149-91-7), chlorogenic (CAS no.: 327-97-9), caffeic (CAS no.: 331-39-5), ferulic (CAS no.: 537-98-4), flavonoids: apigenin (CAS no.: 520-36-5), kaempferol (CAS no.: 520-18-3), luteolin (CAS no.: 491-70-3), myricetin (CAS no.: 529-44-2), quercetin-3-*O*-glucoside (CAS no.: 96862-01-0), anthocyanins: cyanidin-3-*O*-glucoside (CAS no.: 7084-24-4), delphinidin-3-*O*-glucoside (CAS no.: 6906-38-3), malvidin-3-*O*-glucoside (CAS no.: 7228-78-6), pelargonidin-3-*O*-glucoside (CAS no.: 18466-51-8), peonidin-3-*O*-glucoside (CAS no.: 6906-39-4); PBS (phosphate-buffered saline, product no.: P5119).

#### 2.1.2. Origins of Flowers

For experimental purposes, two *Robinia* species were used: *Robinia pseudoacacia* L (black locust) and *Robinia hispida* (bristly locust). The first species is characterized by white flowers (Photo 1, Appendix A), the second species has pink flowers (Photo 2, Appendix A). The current experiment was conducted over two years, 2018–2019. Flowers of both species were collected from six independent trees located in Warsaw parks and green areas. Localization of trees for black locust: Praque Park (52’25’’ N; 21’03’’ E), Sowiński Park (52’23’’ N; 20’95’’ E), Skaryszewski Park (52’24’’ N; 21’06’’ E), Arkadia Upper Park (52’19’’ N; 21’03’’ E), Fort Bema Park (52’26’’ N; 20’94’’ E), Lazienki Park (52’22’’ N; 21’03’’ E). The localization of trees for bristly locust: Mokotowskie Pole Park (52’21’’ N; 20’99’’ E), Skaryszewski Park (52’24’’ N; 21’06’’ E), Sowiński Park (52’23’’ N; 20’95’’ E), Szczęśliwicki Park (52’21’’ N; 20’96’’ E) (1 and 6 of June each year) was done in the morning, and they were quickly transported to the lab.

#### 2.1.3. Plant Material Preparation

Inflorescences were gently divided into single flowers. From each tree, more or less 200–300 hundred individual flowers were collected. The fresh weight of the samples was between 350 and 400 g per tree. Each species sample was divided into two parts. The first part was used for dry matter evaluation, and the second part was freeze-dried using a Labconco (2.5) freeze-dryer (Warsaw, Poland, −40 °C, pressure 0.100 mbar). After freeze-drying, the plant material was ground in a laboratory mill (A-11). Then, the ground samples were stored at −80 °C.

### 2.2. Chemical Analysis

#### 2.2.1. Evaluation of Dry Matter

The dry matter content of the *Robina* flowers was measured before freeze-drying. The dry matter content was determined using the scale method described by Polish Norm PN-R-04013:1988 [22]. Flower samples were dried in 105 °C for 48 h using an FP-25W Farma Play dryer (Bytom, Poland). The dry matter content was calculated for the *Robinia* flower samples based on their mass differences and are given in units of g/100 g FW (fresh weight).

#### 2.2.2. Polyphenols Analysis

##### Sample Preparation

Polyphenols (flavonols and phenolic acids) were measured by the HPLC method [23]. One hundred milligrams of freeze-dried flower powder was mixed with 5 mL of 80% methanol (HPLC-grade) and shaken on a Vortex 326 M (Marki, Poland). Next, all samples were extracted in an ultrasonic bath (10 min, 30 °C, 5.5 kHz). After 10 min of extraction, the flower samples were moved to a centrifuge (10 min, 6000 rpm, 5 °C). After centrifugation, the supernatant was collected in a clean Eppendorf tube and centrifuged again (5 min, 12,000 rpm, 0 °C). A total of 500 µL of supernatant was transferred to HPLC vials and analyzed. 

Polyphenols (anthocyanins) were measured by an HPLC method [24]. The samples were extracted with a mixture of methanol and ultra-pure water (80:20). After the first centrifugation (see previous section), 2.5 mL of the supernatant was collected into a new plastic tube, and then 2.5 mL of 10 M hydrochloric acid (HCl) and 5 mL of pure methanol were added. The samples were gently shaken and put in a refrigerator (5 °C, 10 min). Next, 1 mL of each extract was transferred to an HPLC vial and analyzed.

##### HPLC Set and Analysis Parameters Description

For polyphenol compound separation and identification, a Synergi Fusion-RP 80i column 250 × 4.60 mm (Phenomenex, Warsaw, Poland) was used. Analysis was carried out with the use of Shimadzu equipment (Illinois, USA): two LC-20AD pumps, a CBM-20A controller, an SIL-20AC column oven, and a UV/Vis SPD-20 AV spectrometer. The phenolic compounds (flavonols and phenolic acids) were separated under gradient conditions with a flow rate of 1 mL/min. Two gradient phases were used: 10% (*v*:*v*) acetonitrile and ultra-pure water (phase A) and 55% (*v*:*v*) acetonitrile and ultrapure water (phase B). The phases were acidified with ortho-phosphoric acid (pH 3.0). The total time of the analysis was 38 min. The phase-time program was as follows: 1.00–22.99 min, 95% phase A and 5% phase B; 23.00–27.99 min, 50% phase A and 50% phase B; 28.00–28.99 min, 80% phase A and 20% phase B; and 29.00–38.00 min, 95% phase A and 5% phase B. The wavelengths of detection were 250 nm for phenolic acids and 370 nm for flavonols [23].

Anthocyanins were separated under isocratic conditions with a flow rate of 1.5 mL/min. One mobile phase, acetic acid (5%), methanol (HPLC pure), and acetonitrile (HPLC pure) (70:10:20) were used. The analysis time was 10 min with detection at 570 nm. The anthocyanins were identified by using 99.9% pure standards (Sigma-Aldrich, Warsaw, Poland) and the analysis times for the standards. Anthocyanin compounds were identified only for *Robinia hispida* flowers, because *Robinia pseudoacacia* L. is characterized by white flowers without anthocyanin compounds [24].

##### Compounds Identification and Calculation

The phenolic compounds were identified by using 99.9% pure standards (Sigma-Aldrich, Warsaw, Poland) and the retention times for the standards. Standard curves prepared for all phenolic compounds are presented in Appendix A (Appendix A) . Pure phenolic standards were used for standard solution preparation. From each standard solution, five injections have been made. Every time, a chromatographic pick area was determined and calculated on the basis of standard solution concentration. From standards curves, a mathematical equation was prepared. On the base of the dilution coefficient and equation, the concentration of individual compounds was calculated. 

#### 2.2.3. Antioxidant Activity

A total of 250 mg of the freeze-dried plant material was weighed into a plastic tube, and 25 mL of 80% methanol was added. The tube was placed onto a vortex shaker (LP shaker Vortex, Labo Plus, Kraków, Poland) for 60 s at 2000 rpm. Subsequently, the sample was incubated in a shaker incubator (IKA KS 4000 Control, IKA) for 60 min (temperature 30 °C). After incubation, the sample was shaken again on a vortex shaker for 0.5 min and then centrifuged (Centrifuge, MPW-380 R, Warsaw, Poland) at 5 °C and 6000 rpm for 20 min. After centrifugation, the supernatant was collected. In 10 mL glass tubes, the extract solution was tested and measured with a predetermined dilution scheme (0.5–1.5 mL) to which 3.0 mL of ABTS^·+^ cationic solution in PBS was added. Absorbance measurements were taken exactly 6 min after incubation at 21 °C. The absorbance was measured at a wavelength λ = 734 nm using a spectrophotometer (Helios γ, Thermo Scientific, Warsaw, Poland). The obtained measurements were calculated using a special formula including the dilution factor. The final results are expressed as mmol/g FW of TE (Trolox equivalent) [24].

### 2.3. Statistical Evaluation

The results obtained from the chemical analyses were statistically elaborated using Statgraphics Centurion 15.2.11.0 software (StatPoint Technologies, Inc., Warranton, VA, USA). Table 1 presents values expressed as the means for the black locust and bristly locust in both experimental years (n = 12). Moreover, the interaction between the experimental year and species is shown in the table (n = 6). The standard error is presented along with the average value. Statistical calculations were based on two-way analysis of variance with the use of Tukey’s test (*p* < 0.05). A lack of statistically significant differences between the examined groups is indicated by labeling with the same letters. Principal component analysis (PCA) was performed using XLSTAT software (XLSTAT, 2020, New York, NY, USA) to categorize the farming systems used in 2018–2019 and individual species examined in both experimental years based on their bioactive compound contents.

## 3. Results and Discussion

The results from the obtained analysis are presented in Table 1. The results of individual phenolic compounds are presents in Figure 1, Figure 2 and Figure 3, and compound spectra are presented in Figure 4. In both years of the experiment, bristly locust flowers contained more dry matter, but there was no significant difference. Our observations showed that *Robinia hispida* flowers are a high dry matter content species compared to *Robinia pseudoacacia* flowers. Different concentrations of dry matter in flowers are an effect of the genetic diversification of species. As reported by others, the dry matter content among 11 edible flower species was in the range of 3.75–34.01 g/100 g FW (for *Begonia × tuberhybrida* Voss. and *Lavandula angustifolia* Mill.) [25]. In the present experiment, we observed that bristly locust flowers (pink) contained significantly more dry matter than black locust flowers (white), with values of 15.72 g/100 g FW and 14.03 g/100 g FW, respectively.

A similar relationship was observed with daisy flowers. Pink daisy flower species contained 18.63 g/100 g FW and white daisy flowers contained 15.97 g/100 g FW [25]. Polyphenols are one of the most important and pro-healthy secondary metabolites produced by plants. They function in plant flowers to make the flowers more attractive to pollinating insects [26]. Polyphenols are connected with floral nectar and pollen production as well as color recognition by pollinator insects [27]. In the present experiment, we observed significantly more total polyphenols in the pink flower species (bristly locust) compared to the white flower species (black locust) (*p* < 0.0001) in both experimental years (*p* = 0.0017). Similar results were observed with heathier flowers. Bright flower Erica species contained half of the total polyphenols (42.9 mg/g DW, dry weight) compared to pink flower *Erica* species (81.0 mg/g DW) [28]. Daisies with white flowers contained 0.31 mg/g FW of total polyphenols and pink daisies contained 0.44 mg/g FW [25]. In both years of the experiment, greater and significant total phenolic acid (*p* = 0.005 and *p* < 0.0001) concentrations were observed in bristly locust flowers compared to black locust flowers (Table 1). A similar situation was observed by Cunja et al. (2014) with different rose flowers. Roses with pink flowers contained 0.57 mg/g FW total phenolic acids, whereas white flowers contained only 0.27 mg/g FW [29]. Four individual phenolic acids were identified in *Robinia* species flowers: gallic, chlorogenic, caffeic, and ferulic acids (Figure 1).

The presence of phenolic acids in *Robinia* flowers is connected with their occurrence in flower nectar and honey. Characteristic phenolic acid concentrations and profiles in black locust honey were examined by others [30]. In the present experiment, among the different phenolic acids, two of them, ferulic acid and caffeic acid, were identified (Table 1). Only significant differences between *Robinia* flowers were observed in the case of caffeic acid concentration. Bristly locust flowers contained significantly more caffeic acid (*p* < 0.0001) than black locust flowers. As reported by others, black locust flowers contain 0.77 mg/g of FW gallic acid. In our experiment, we noted that the examined species affected the concentration of gallic acid in flowers [31]. Bristly locust flowers contained 0.75 mg/g FW, and black locust flowers contained only 0.35 mg/g FW gallic acid (Table 1). Chlorogenic acid was identified in black locust flowers at a concentration of 1.48 mg/g FW, and bristly locust flowers had a concentration of 1.90 mg/g FW. A similar result was reported by Gayibova et al. (2019) [32]. Flavonoids are the next most important pro-healthy bioactive compounds in *Robinia* species flowers. The flavonoids group is very huge, so these compounds have been divided into additional subgroups with different chemical structure and character. The compounds belong to flavonoids as well purple anthocyanins. Therefore, it is quite obvious that purple and pink flowers could contain a higher concentration of flavonoids than white flowers [9,33]. Moreover, flowers of different colors (white and pink) present a different concentration of the other flavonoid compounds. In our experiment, white black locust flowers contained significantly more total flavonols (*p* = 0.005) compared to pink bristly locust flowers, with values of 1.74 mg/g and 1.68 mg/g FW. Similar results with different rose flowers were presented by Cunja et al. (2014) [34]. White rose flowers contained 4.56 mg/g and pink flowers contained only 2.23 mg/g FW total flavonols. However, it should be emphasized that this phenomenon does not occur in every species. White *Magnolia* flowers contain 11.71 mg/g DW flavonols, but pink flowers contain 17.31 mg/g DW total flavonols [35]. A large proportion of higher plants have white, ivory, or cream flowers. Surveys of white-flowering wild and cultivated plant species by Deng and Bhattarai (2018) showed that the flavonols luteolin, apigenin, and myricetin commonly occur in white flower petals [36]. Our findings confirmed this phenomenon in *Robinia* flowers. Black locust flowers contained significantly more luteolin (*p* = 0.006) and myricetin (*p* = 0.0002) than bristly locust flowers (Table 1). Kaempferol was the last flavonol compound identified in the present experiment (Figure 2).

*Robinia* with pink flowers was characterized by a 3-fold higher concentration of kaempferol compared to the white flower species. A similar finding was presented by Mato et al. (2001), with white and light pink as well deep pink carnation flowers [36]. They observed that more deep pink flowers also contained more kaempferol. Only pink, purple, and violet flowers contained anthocyanins. These plant pigments create characteristic flower colors. Bristly locusts have flowers with a pink color. Five different anthocyanins were identified in the flowers of this species (Figure 3). All anthocyanins occur in their glucoside form. The main two anthocyanins with the highest concentrations were cyanidin-3-*O*-glucoside and pelargonidin-3-*O*-glucoside (Table 1). Anthocyanins have strong antioxidant activity, and together with other identified flavonoid compounds, they create the total biological and antioxidant power of locust flowers. In the present experiment, bristly locust flowers showed significant (*p* < 0.0001) 3-fold higher antioxidant power compared to black locust flowers (Table 1). On the other hand, bioactive flavonol group compounds have antioxidant properties as well. We found a high correlation score between total polyphenols and antioxidant capacity in bristly and black locust flowers (Table 1). Even though anthocyanins were only found in pink locust flowers, other polyphenols create the antioxidant power of black locust flowers. PCA analysis showed a high and significant overall variation in both years of the experiment. Overall, 95.42% was explained by PC1 and PC2 (Figure 5). The degree of dependence between the black and bristly locust flowers and the factors marked as total flavonoids (TFv), ferulic acid (FA), myricetin (My), and luteolin (Lut) contents were particularly important in both experimental years for black locust flowers. The rest of the experimental factors were important for bristly locust flowers. It is worth pointing out that both *Robinia* species were located in different sections of the graph and were mostly dependent on different polyphenol compounds.

## 4. Conclusions

The present experiment confirms that *Robinia* flowers contain different qualities and quantities of polyphenols. On the other hand, the two-year experiment provides more specific and accurate answers to questions about the profile and concentration of phenolics. It is worth pointing out that the vast majority of the research presented in the literature on polyphenols in *Robinia* flowers focused only on black and not bristly locust flowers. Due to the high concentration of various bioactive compounds, locust flowers can become very good food additives with high biological value.

## Figures and Tables

**Figure 1 biomolecules-10-01603-f001:**
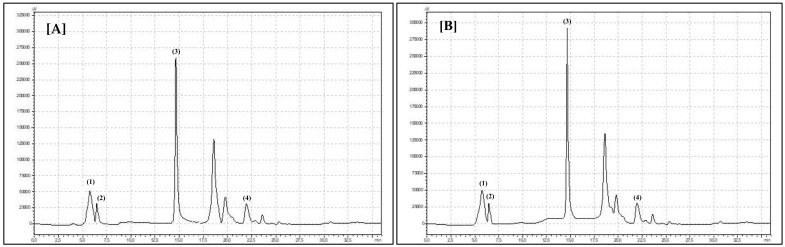
Chromatogram showing retention times for phenolic acids identified in flowers of (**A**) *Robinia pseudoacacia* L. and (**B**) *Robinia hispida*: (1) gallic acid, (2) chlorogenic acid, (3) caffeic acid, and (4) ferulic acid.

**Figure 2 biomolecules-10-01603-f002:**
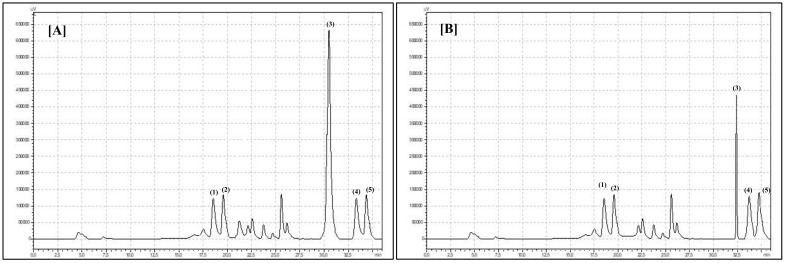
Chromatogram showing retention times for flavonols identified in flowers of (**A**) *Robinia pseudoacacia* L. and (**B**) *Robinia hispida*: (1) myricetin, (2) apigenin, (3) luteolin, (4) quercetin-3-*O*-glucoside, and (5) kaempferol.

**Figure 3 biomolecules-10-01603-f003:**
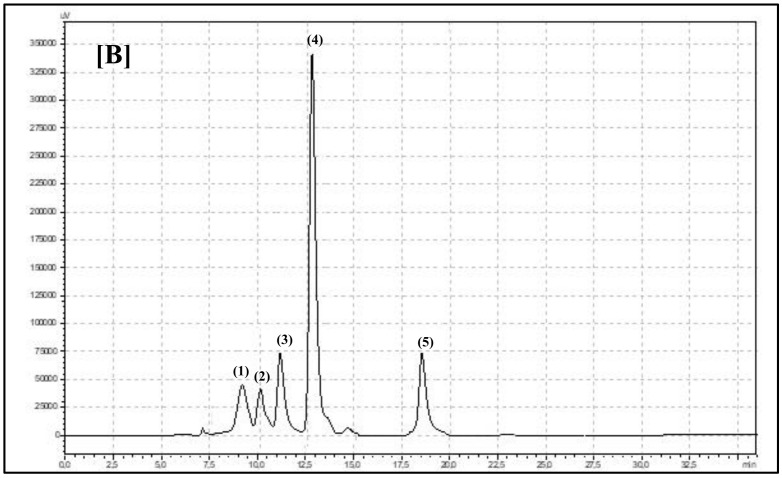
Chromatogram showing retention times for anthocyanins identified in flowers of (B) *Robinia hispida*: (1) malvidin-3-*O*-glucoside, (2) peonidin-3-*O*-glucoside, (3) cyanidin-3-*O*-glucoside, (4) pelargonidin-3-*O*-glucoside, and (5) delphinidin-3-*O*-glucoside.

**Figure 4 biomolecules-10-01603-f004:**
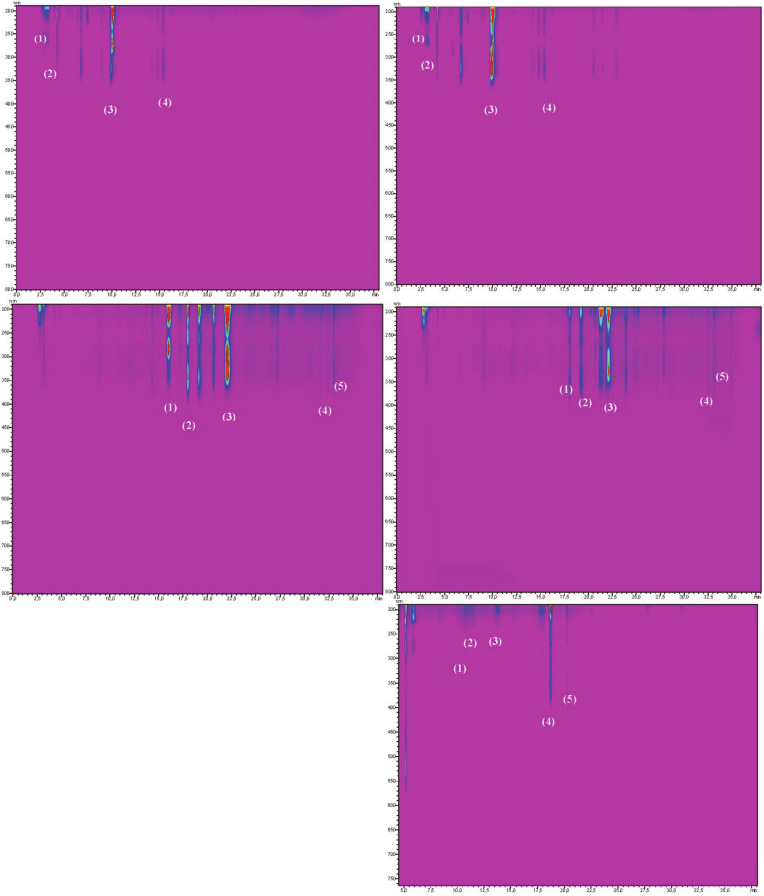
Picture showing spectra area and retention times in black locust [A] and bristly locust [B] for phenolic acids [A1] and [B1]; flavonols [A2] and [B2]; and anthocyanins [B3].

**Figure 5 biomolecules-10-01603-f005:**
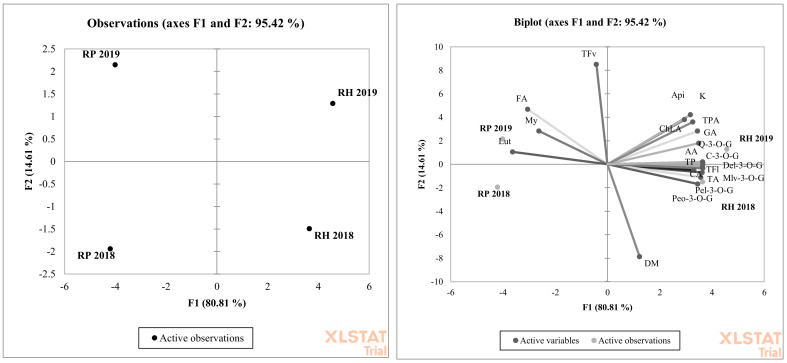
Principal component analysis (PCA) showing the relationship between the polyphenols composition and *Robinia* species: *Robinia pseudoacacia* L (RP), *Robinia hispida* (RH) in two years (2018 and 2019) of experiment. Dry matter (DM,) total polyphenols (TP), total phenolic acids (TPA), gallic acid (GA), caffeic acid (CA), ferulic acid (FA), chlorogenic acid (ChLA), total flavonoids (TF), total flavonols (TFv), quercetin-3-*O*-glucoside (Q-3-*O*-G), myricetin (My), apigenin (Api), kaempferol (K), total anthocyanins (TA), cyanidin-3-*O*-glucoside (Cy-3-*O*-G), pelargonidin-3-*O*-glucoside (Pel-3-*O*-G), malvidin-3-*O*-glucoside (Mlv-3-*O*-G), peonidin-3-*O*-glucoside (Peo-3-*O*-G).

**Table 1 biomolecules-10-01603-t001:** Concentration of dry matter (in g/100 g FW), phenolic compounds (in mg/g FW), and antioxidant activity (in mM/g FW) in examined *Robinia* flowers in two years of experiment (2018–2019).

Species and Experimental Year/Bioactive Compounds	2018		2019		*Robinia pseudoacacia* L. (White Folwers)	*Robinia hispida* (Pink Flowers)	*p*-Value	
	*Robinia pseudoacacia* L. (White Folwers)	*Robinia hispida* (Pink Flowers)	*Robinia pseudoacacia* L. (White Folwers)	*Robinia hispida*(Pink Flowers)			Year x Species	Species
dry matter	15.88 ± 0.17 a	17.71 ± 0.04 a	12.17 ± 0.15 a	13.72 ± 0.04 a	14.03 ± 0.55 B	15.72 ± 0.57 A	N.S.	<0.0001
total polyphenols	3.51 ± 0.01 b	10.81 ± 0.13 a	4.31 ± 0.06 b	11.08 ± 0.07 a	3.91 ± 0.12 B	10.94 ± 0.08 A	0.0017	<0.0001
total phenolic acids	1.94 ± 0.01 c	2.94 ± 0.03 b	2.39 ± 0.07 b	3.27 ± 0,02 a	2.16 ± 0.07 B	3.10 ± 0.05 A	0.005	<0.0001
gallic acid	0.37 ± 0.01 c	0.66 ± 0.12 b	0.41 ± 0.01 c	0.85 ± 0.08 a	0.39 ± 0.05 B	0.75 ± 0.07 A	<0.0001	<0.0001
chlorogenic acid	1.25 ± 0.01 c	1.90 ± 0.02 a	1.70 ± 0.01 b	1.90 ± 0.07 a	1.48 ± 0.05 B	1.90 ± 0.06 A	<0.0001	<0.0001
caffeic acid	0.25 ± 0.00 b	0.34 ± 0.13 ab	0.16 ± 0.00 c	0.47 ± 0.05 a	0.20 ± 0.00 B	0.40 ± 0.09 A	<0.0001	<0.0001
ferulic acid	0.07 ± 0.01 a	0.04 ± 0.01 a	0.11 ± 0.00 a	0.05 ± 0.01 a	0.09 ± 0.01 A	0.04 ± 0.03 A	N.S.	N.S.
total flavonoids	1.56 ± 0.01 c	7.88 ± 0.02 a	1.92 ± 0.01 b	7.81 ± 0.01 a	1.74 ± 0.07 B	7.84 ± 0.01 A	0.005	<0.0001
total flavonols	1.56 ± 0.00 a	1.54 ± 0.01 a	1.92 ± 0.00 a	1.83 ± 0.01 a	1.74 ± 0.01 A	1.68 ± 0.02 A	N.S.	N.S.
quercetin-3-*O*-glucoside	0.06 ± 0.00 a	0.37 ± 0.00 a	0.08 ± 0.06 a	0.40 ± 0.00 a	0.07 ± 0.03 B	0.38 ± 0.00 A	N.S.	<0.0001
myricetin	0.39 ± 0.00 a	0.39 ± 0.02 a	0.43 ± 0.00 a	0.36 ± 0.07 b	0.41 ± 0.00 A	0.37 ± 0.03 B	0.0002	<0.0001
luteolin	0.77 ± 0.01 a	0.04 ± 0.01 b	0.84 ± 0.01 a	0.07 ± 0.01 b	0.80 ± 0.01 A	0.05 ± 0.01 B	0.006	<0.0001
apigenin	0.24 ± 0.01 b	0.39 ± 0.00 a	0.34 ± 0.01 ab	0.46 ± 0.00 a	0.29 ± 0.01 B	0.42 ± 0.00 A	0.012	<0.0001
kaempferol	0.11 ± 0.00 c	0.35 ± 0.01 b	0.24 ± 0.01 c	0.55 ± 0.00 a	0.17 ± 0.02 B	0.45 ± 0.01 A	<0.0001	<0.0001
total anthocyanins	N.D.	6.34 ± 0.00 a	N.D.	5.98 ± 0.00 b	N.D.	6.16 ± 0.03	0.009	<0.0001
cyanidin-3-*O*-glucoside	N.D.	1.52 ± 0.02 b	N.D.	1.87 ± 0.01 a	N.D.	1.70 ± 0.05	<0.0001	<0.0001
pelargonidin-3-*O*-glucoside	N.D.	4.41 ± 0.13 a	N.D.	3.68 ± 0.06 b	N.D.	4.05 ± 0.14	<0.0001	<0.0001
delphinidin-3-*O*-glucoside	N.D.	0.21 ± 0.01 a	N.D.	0.25 ± 0.01 a	N.D.	0.23 ± 0.01	N.S.	<0.0001
malvidin-3-*O*-glucoside	N.D.	0.15 ± 0.00 a	N.D.	0.16 ± 0.00 a	N.D.	0.15 ± 0.00	N.S.	<0.0001
peonidin-3-*O*-glucoside	N.D.	0.03 ± 0.00 a	N.D.	0.02 ± 0.00 a	N.D.	0.03 ± 0.00	N.S.	<0.0001
antioxidant activity	1.43 ± 0.01 b	4.40 ± 0.05 a	1.75 ± 0.03 b	4.51 ± 0.03 a	1.59 ± 0.05 B	4.45 ± 0.03 A	0.0017	<0.0001

Data are presented as the mean ± SE with ANOVA *p*-value; Means in rows followed by the same letter are not significantly different at the 5% level of probability (*p* < 0.05); N.S. not significant statistically; N.D. compound not detected; number of replications (n) = 6.

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
