# Peer review of "Quantitative and Qualitative Identification of Bioactive Compounds in Edible Flowers of Black and Bristly Locust and Their Antioxidant Activity"

_biomolecules, 2020, doi:10.3390/biom10121603_

Round 1

Reviewer 1 Report

This research article proposes a simple protocol to extract metabolites from Black and bristly locust flowers, and their characterization by means of HPLC. Several research articles were published on this topic.

Introduction section is very concise; author should take into consideration a larger number of already published research articles on this topic. In my opinion, the novelty presented should be more stressed. Experimental part is a little bit confusing. Some information (like chromatograms and UV spectra) should be moved in results section.  Moreover, my suggestion is to describe more schematically the experimental design: 1) Materials and samples; 2) Sample extraction; 3) HPLC instrumentation; 4) HPLC methods; 5) Antioxidant activity; 6) Statistical evaluation. Sample extraction recovery methods (for both polyphenols and anthocyanins) must be developed. How the author quantified the bioactive molecules content present in Table 1? No reagents were described in experimental section, and no method validation was presented. All these information are missing.

Chromatographic separation proposed in figures 1-3 showed that the peak identified and quantified have not a Gaussian shape. Maybe there is something to improve in the separation method or in the instrumental parameters. My suggestion is to repeat the LC-UV analyses in order to improve the peak shape.

Minor revision

- Paragraph 2.1. Could the authors explain why picked the flowers between the 5 and 6 of June? And why only in the morning?

- Paragraphs 2.1-2.3. Could the authors give in the text information on: 1) number of flowers used for the experimental design; 2) the weight of flowers; 3) the weight of dry matter.

Author Response

Reviewer no. 1

Thank you very much for the review and for your suggestions and opinions about my manuscript in the “Biomolecules” journal. All pointed comments were put into the text. All comments reply are presented below

Comment 1: “…. In my opinion, the novelty presented should be more stressed..…”

Authors’ response: As suggested by the Reviewer, the study (Introduction section, the aim of the study) underlined the needing for such research

Comment 2: “…. Experimental part is a little bit confusing. Some information (like chromatograms and UV spectra) should be moved in results section…. “

Authors’ response: As suggested by the Reviewer all Figures with chromatograms and spectra’s were transferred to Result section.

Comment 3: “…Moreover, my suggestion is to describe more schematically the experimental design: 1) Materials and samples; 2) Sample extraction; 3) HPLC instrumentation; 4) HPLC methods; 5) Antioxidant activity; 6) Statistical evaluation….”

Authors’ response: According to Reviewer suggestion all Material and methods section was changed, as well the new sub-sections were created. This make this section better organized and easier for read.

Comment 4: “…How the author quantified the bioactive molecules content present in Table 1? No reagents were described in experimental section, and no method validation was presented. All these information are missing….”

Authors’ response: All missing information about the method of identification of compounds and calculation methods, standard curves are included in the sub-section: “Compounds identification and calculation”. All prepared and used standard curves prepared and presented as Supplementary materials (Figures S1-S3).

Comment 5: “…Chromatographic separation proposed in figures 1-3 showed that the peak identified and quantified have not a Gaussian shape. Maybe there is something to improve in the separation method or in the instrumental parameters. My suggestion is to repeat the LC-UV analyses in order to improve the peak shape…..”

Authors’ response: Author agree with Reviewer remark. Analytic material was extracted with all detail and described conditions. Author use described polyphenols method many times and obtain similar, sometime better chromatogram picks shape. Prepared chromatogram of standards  were in similar shape, so Author accepted shape as with good quality. Of course Author pay attention for Reviewer remark and in the future another experiment with validation of different method for polyphenols extraction and analysis condition will be prepared. This activity is important to obtain the best results.

Comment 6: “…Paragraph 2.1. Could the authors explain why picked the flowers between the 5 and 6 of June? And why only in the morning?...”

Authors’ response: Flowers (Flores), according to the information contained in the Polish Pharmacopoeia, the optimal time of harvesting flowers is the beginning of flowering of plants, when they reach the highest level of vitality and contain the most active ingredients. Inflorescentia, (i.e. clusters of flowers) on a plant collected as a whole, sometimes used interchangeably with flowers, e.g. robinia inflorescence. In 2018, Robinia trees flowering started on June 5, and in 2019 it was on June 6. The best time to harvest flowers (herbal material) is in the early morning hours, when the flowers have time to dry from the dew.

Comment 7: “…- Paragraphs 2.1-2.3. Could the authors give in the text information on: 1) number of flowers used for the experimental design; 2) the weight of flowers; 3) the weight of dry matter….”

Authors’ response: According to Reviewer suggestion in sub-section 2.1.3. Plant material preparation  Author add information about number of flowers, fresh weight of flowers. Dry matter of 100 g of flowers is presented in Table 1. 

Reviewer 2 Report

The article “Quantitative and qualitative identification of bioactive compounds in edible flowers of black and bristly locust and their antioxidant activity” describes the phenolic and anthocyanin contents of two related species over two years.

Even though the current study would be of interest due to the limited information in the literature regarding the bioactive compounds in bristly locusts, there are several major concerns.

The author evaluated the presence and the quantities of several polyphenols present in both species, however it should be noted that the number of compounds evaluated is rather small (4 compounds). I appreciate that the author used a HPLC method for a more precise determination of the bioactive compounds, however I would suggest to add also a total polyphenolic content determination to circumvent the limited number of compounds quantified with the HPLC method.

Another major technical drawback of the current study is related to the correlations that were established between bioactive compounds quantified with the HPLC methods and the antioxidant assay used. For the quantifications the authors used an ultrasound-assisted extraction with a MeOH:H2O mixture while for the antioxidant assay an extraction in water at 30°C. Due to the differences in the extraction methods no correlation can be established as the aqueous extract could be qualitatively and quantitatively different. I suggest if a correlation is desired, the authors should use the same extraction technique. The lack of correlations derived from PCA is most probably the result of the above-mentioned issue.

The manuscript is poorly written, and the writing could be better. The article presents a lot of short sentences that are not properly linked. Several exemples are:

lines 26-27 “In Poland, they are planted mostly as ornamental trees that have white and pink flowers, but are useful and also grow in the wild.”

lines 30-31 “In addition to fruits and vegetables, the flowers are also consumable”

lines 173 “Flavonoids are chemical compounds from different chemical groups”

Some of the conclusions drawn by the author don’t seem to have a scientific rationale      ( “notably, the color of the flowers can affect the dry matter content”).

If the authors can not provide more assays for the characterization of the bioactive compounds in these two species, I suggest do add more extraction conditions to evaluate which one is the most efficient in extracting the described bioactive compounds.

Overall due to the limited number of assays, the technical issues and the writing that would need to be improved, I recommend that the present article should be rejected.

Author Response

Reviewer no. 2

Thank you very much for the review and for your suggestions and opinions about my manuscript in the “Biomolecules” journal. All comments replies are presented below. All corrections were made in manuscript text.

Comment 1: “…. Even though the current study would be of interest due to the limited information in the literature regarding the bioactive compounds in bristly locusts, there are several major concerns.

The author evaluated the presence and the quantities of several polyphenols present in both species, however it should be noted that the number of compounds evaluated is rather small (4 compounds). I appreciate that the author used a HPLC method for a more precise determination of the bioactive compounds, however I would suggest to add also a total polyphenolic content determination to circumvent the limited number of compounds quantified with the HPLC method...…”

Authors’ response: With all respect to Reviewer and according to data presented in Table 1 not 4, but 9 in total polyphenolic compounds (4 from phenolic acids group and 5 from flavonoids group) have been found in black locust flowers as well 13 compounds identified in bristly locust flowers (4 from phenolic acids group and 5 from flavonoids group, 4 from anthocyanins group). In the present literature in similar published already references not more than 4 phenolic compounds were presented by another scientists. In many articles only total polyphenols by Folin-Ciocalteu method were measured. In that light Authors claim, that identification of 9 and 13 polyphenolic compounds is an success. Below is presented the list of references showing similar experiments:

Savic Gajic, I.; Savic, I.; Boskov, I.; Žeraji´c, S.; Markovic, I.; Gajic, D. Optimization of ultrasound-assisted extraction of phenolic compounds from black locust (Robiniae Pseudoacaciae) flowers and comparison with conventional methods. Antioxidants 2019, 8, 248, 1-14

rutin, epigallocatechin, ferulic acid, quercetin

XiaoYang, W.,  Lin, T., Lei, Z.,  YanNi, L., ZhenYan, Z. Determination of polyphenols in flowers of Robinia pseudoacacia L. by Folin-Ciocaileu method, Food and Drug, 2010, 12  332-335

only total polyphenols by Folin-Ciocalteu method

Kowalczewski, P. Ł., Pauter, P., Smarzyński, K., Różańska, M. B., Jeżowski, P., Dwiecki, K., & Mildner‐Szkudlarz, S. Thermal processing of pasta enriched with black locust flowers affect quality, phenolics, and antioxidant activity, J. Food Proc. Pres., 2019, 43, 14106, 1-11.

kaempferol-3-O-glucoside (5 isomers), gallic acid and total polyphenols by Folin-Ciocalteu method

Hefeng,S.Y.L.J.X. Study on the chemical composition of Robinia pseudoacacia flowers, Chem. Ind. Forest Prod., 1992, 4, 1-5

only total polyphenols by Folin-Ciocalteu method

Stankov,  S., Fidan, H., Ivanova, T., Stoyanova, A., Damyanova, S., Desyk M. Chemical composition and application of flowers of false acacia (Robinia pseudoacacia L.) Ukrainian Food J., 2018, 7, 4, 877-887.

only total polyphenols by Folin-Ciocalteu method

Sarikurkcu, C., Kocak, M. S., Tepe, B., & Uren, M. C.  An alternative antioxidative and enzyme inhibitory agent from Turkey: Robinia pseudoacacia L. Industrial Crops and Products, 2015, 78, 110–115.

only total polyphenols by Folin-Ciocalteu method

Comment 2: “….Another major technical drawback of the current study is related to the correlations that were established between bioactive compounds quantified with the HPLC methods and the antioxidant assay used. For the quantifications the authors used an ultrasound-assisted extraction with a MeOH:H2O mixture while for the antioxidant assay an extraction in water at 30°C. Due to the differences in the extraction methods no correlation can be established as the aqueous extract could be qualitatively and quantitatively different. I suggest if a correlation is desired, the authors should use the same extraction technique. The lack of correlations derived from PCA is most probably the result of the above-mentioned issue.…. “

Authors’ response: The author wants to apologize for the mistake. After careful examination of the analysis protocol, an error was found in the description of the solvent used to measure the antioxidant activity. Of course Author agree with Reviewer remark, that should be use a the same solvent.  That why is possible to  make correlation of both factors. According to rules for both analysis 80% of methanol have been used. Pointed mistake was corrected into manuscript text.

Comment 3: “…lines 26-27 “In Poland, they are planted mostly as ornamental trees that have white and pink flowers, but are useful and also grow in the wild.”….”

Authors’ response: Authors agree with Reviewer remark. Poorly written sentence was corrected into manuscript text.

Comment 4: “…lines 30-31 “In addition to fruits and vegetables, the flowers are also consumable”….”

Authors’ response: Authors agree with Reviewer remark. Poorly written sentence was corrected into manuscript text.

Comment 5: “…lines 173 “Flavonoids are chemical compounds from different chemical groups”…”

Authors’ response: Authors agree with Reviewer remark. Poorly written sentence was corrected into manuscript text.

Comment 6: “…Some of the conclusions drawn by the author don’t seem to have a scientific rationale      ( “notably, the color of the flowers can affect the dry matter content”).…”

Authors’ response: The author wants to apologize for a improperly constructed sentence in the manuscript text. The sentence was removed.

Comment 7: “…If the authors can’t provide more assays for the characterization of the bioactive compounds in these two species, I suggest do add more extraction conditions to evaluate which one is the most efficient in extracting the described bioactive compounds.…”

Authors’ response: With all due respect to the Reviewer, the presented experiment is not an improvement of the methodology, but a demonstration of differences in the composition of polyphenolic compounds in two Robinia species. Of course, the Reviewer's remark is very valuable and will be used in the future in the new, planned experiment of validating polyphenol extraction methods. On the other hand many experiment founded in literature show such validation methods, non of them comparison of phenolic compounds among to Robinia species with detail compounds identification in more than one year. In presented experiment 9 of polyphenols compounds in black locust flower and 13 polyphenols in bristly locust flower have been identified. The Author of the presented manuscript believes that this is a sufficient number of identified compounds. Especially that in other presented works there are significantly less of these compounds.

Reviewer 3 Report

Manuscript entitled “Quantitative and qualitative identification of bioactive compounds in edible flowers of black and bristly locust and their antioxidant activity” is recommended to be published in Biomolecules after minor revision.

Please consider following suggestions and corrections:

Line 19: myricetin instead of myrycetin

Lines 44, 47, 59, 64: Robinia in italic

Line 47: Robinia pseudoacacia in italic

Line 48: Robinia hispida in italic

Lines 48 and 49: Photo 1 and Photo 2 are not present in the manuscript

Line 51: Can you pinpoint the habitat for both species in Warsaw? Can you give coordinates?

Lines 56, 57, 63, 69, 70, 72, 93, 105, 107, and 110: insert space between a number and a unit of measurement for temperature e.g. -40 °C

Line 73: 250 × 4.60 with spaces

Line 86: Verb “are” is missing, and full stop at the end of the sentence is missing.

Lines 99 and 100: Latin names in italic: Robinia hispida and Robinia pseudoacacia

Line 109: ABTS+ with dot and plus in superscript

Line 118: n = 12 with spaces

Line 120: p < 0.05 with spaces and p in italic

Lines 128 and 129: Latin names in italic: Robinia hispida and Robinia pseudoacacia

Line 134: 100 g with space

Table 1: myricetin instead of myrycetin

Line 135: p-value, p < 0.05 with p in italic

Lines 143, 147, 166: with spaces and p in italic e.g. (p < 0.0001), (p = 0.0017)

Lines 144 and 145: Erica in italic

Line 149: add reference [30]

Line 160: add "acid" after "ferulic"

Lines 161, 164, 172, 186, 198, 218, 246, 249: Robinia in italic

Lines 178, 187, 208: with spaces and p in italic e.g. (p < 0.0001), (p = 0.0017)

Line 181: Magnolia in italic

Line 185, 197, 216, 243: myricetin instead of myrycetin

Line 226: move numbers above the peaks

Line 230: pelargonidin instead of pelargonidyn

Line 240: omit analysis after (PCA)

Lines 243 and 244: quercetin-3-O-glucoside, cyanidin-3-O-glucoside, pelargonidin-3-O-glucoside, malvidin-3-O-glucoside, peonidin-3-O-glucoside with O in italic

Line 255: I recommend omitting “All authors have read and agreed to the published version of the manuscript.” since there is only one author.

Author Response

Reviewer no. 3

Thank you very much for the review and for your suggestions and opinions about my manuscript in the “Biomolecules” journal.

Comment 1: “…Line 19: myricetin instead of myrycetin.….”

Authors’ response: According to Reviewer suggestion pointed error has been corrected in the all places in the manuscript text.

Comment 2: “….Lines 44, 47, 59, 64: Robinia in italic, Line 47: Robinia pseudoacacia in italic

Line 48: Robinia hispida in italic…”

Authors’ response: According to Reviewer suggestion (italic type) instead normal type in latina names of examined species, errors has been corrected in the all places in the manuscript text.

Comment 3: “….Lines 48 and 49: Photo 1 and Photo 2 are not present in the manuscript….”

Authors’ response:

Comment 4: “…Lines 48 and 49: Photo 1 and Photo 2 are not present in the manuscript.….”

Authors’ response: Photos of examined Robinia flowers were posted in Supplementary materials. The information about the localization of  photos was added in sub-section 1.2.1. Origins of flowers,  line 64-65

Comment 5: “…Line 51: Can you pinpoint the habitat for both species in Warsaw? Can you give coordinates?.….”

Authors’ response: According to Reviewer suggestion park names and geographical coordinates where flowers of examined species were collected were added in manuscript text, sub-section: 1.2.1. Origins of flowers, lines: 68-70.

Comment 6: “….Lines 56, 57, 63, 69, 70, 72, 93, 105, 107, and 110: insert space between a number and a unit of measurement for temperature e.g. -40 °C….”

Authors’ response: According to Reviewer suggestion pointed error has been corrected in the all places in the manuscript text.

Comment 7: “…Line 73: 250 × 4.60 with spaces.….”

Authors’ response: According to Reviewer suggestion pointed error has been corrected in the all places in the manuscript text.

Comment 8: “…Line 86: Verb “are” is missing, and full stop at the end of the sentence is missing..….”

Authors’ response: According to Reviewer no 1,  this sentence was re-located to Results section. The pointed error was corrected into manuscript text.

Comment 9: “…Line 109: ABTS•+ with dot and plus in superscript.….”

Authors’ response: According to Reviewer suggestion pointed error has been corrected in the manuscript text.

Comment 10: “….Line 118: n = 12 with spaces….”, Line 120: p < 0.05 with spaces and p in italic,

Authors’ response: According to Reviewer suggestion pointed errors have been corrected in the manuscript text.

Comment 11: “….Line 134: 100 g with space….”

Authors’ response: According to Reviewer suggestion pointed error has been corrected in the manuscript text.

Comment 12: “…Line 135: p-value, p < 0.05 with p in italic, Lines 143, 147, 166: with spaces and p in italic e.g. (p < 0.0001), (p = 0.0017).….”

Authors’ response: According to Reviewer suggestion pointed errors have been corrected in the manuscript text.

Comment 13: “…Line 149: add reference [30].….”

Authors’ response: As suggested by the Reviewer no. 1, several literature items were added, in the pointed place reference no. 33 was given.

Comment 14: “….Line 160: add "acid" after "ferulic"….”

Authors’ response: According to Reviewer suggestion pointed error has been corrected in the manuscript text.

Comment 15: “…Line 181: Magnolia in italic.….”

Authors’ response: According to Reviewer suggestion pointed error has been corrected in the manuscript text.

Comment 16: “….Line 226: move numbers above the peaks….”

Authors’ response: According to Reviewer suggestion numbers were moved above the picks. I must to explain, that numbers are relocated in time of manuscript submission.

Comment 17: “….Line 230: pelargonidin instead of pelargonidyn….”

Authors’ response: According to Reviewer suggestion pointed error has been corrected in the manuscript text.

Comment 18: “…Line 240: omit analysis after (PCA).….”

Authors’ response: According to Reviewer suggestion pointed error has been corrected in the manuscript text.

Comment 19: “….Lines 243 and 244: quercetin-3-O-glucoside, cyanidin-3-O-glucoside, pelargonidin-3-O-glucoside, malvidin-3-O-glucoside, peonidin-3-O-glucoside with O in italic….”

Authors’ response: According to Reviewer suggestion pointed errors have been corrected in the manuscript text.

Comment 20: “…Line 255: I recommend omitting “All authors have read and agreed to the published version of the manuscript.” since there is only one author..….”

Authors’ response: According to Reviewer suggestionauthors” has been change into “author”

Round 2

Reviewer 1 Report

Dear author,

with the corrections carried out in the revised version of the manuscript the research article is now suitable for publication.

Author Response

Reviewer no. 1

Thank you very much for the review and for your suggestions and opinions about my manuscript in the “Biomolecules” journal. All pointed comments were put into the text. All comments reply are presented below

Comment 1: “…. with the corrections carried out in the revised version of the manuscript the research article is now suitable for publication...…”

Authors’ response: Thank you for your opinion about submitted manuscript. In case of Introduction section small correction have been made (according to suggestion Reviewer no. 2), which is in accordance with suggestion of Reviewer no. 2 (lines 33-39).

Reviewer 2 Report

Based on the answers and the modification done to the original draft, I recommend that version 2 of the article should be accepted after minor revision.

Overall the article is well-written, but I still consider that the introduction part could be better written.

Lines 55-59. Should be re-written as all three sentences describe the same idea.

Line 39. The idea was presented previously in lines 30-31.

Lines 33-35. The sentences should be better linked.

Suggestion: the small sentences “Yellow and orange colours are created by carotenoids and flavonoids” and “Purple, violet and blue colours are from anthocyanins” can be better written in one sentence “ Depending on the colour of the flowers, different phytochemical profiles were observed, with yellow and orange flowers displaying higher concentrations of carotenoids and flavonoids, while purple, violet and blue coloured flowers displaying higher quantities of anthocyanins.”

Author Response

Reviewer no. 2

Thank you very much for the review and for your suggestions and opinions about my manuscript in the “Biomolecules” journal. All comments replies are presented below. All corrections were made in manuscript text.

Comment 1: “….Lines 55-59. Should be re-written as all three sentences describe the same idea...…”

Authors’ response: Section 2.1.1. Chemical (lines 55-61) was added in the first round of manuscript correction. It was a suggestion of Reviewer no. 1 to add separate listed chemicals used in all analysis. Full names of chemical reagents are in accordance in international chemical standards names. Chemicals were listed in alphabetical order with name of Deliver and country origin (in brackets). Due to the all respect to Reviewer how should I re-write this section? Here is no ideas, but only chemical names, which were carefully checked in case of chemical names correctness. Only one was added. CAS numbers for all chemicals. This help better to identified all used chemicals. Phenolic standard now are as well in alphabetical order within specifically chemical groups belong to. 

Comment 2: “…..Line 39. The idea was presented previously in lines 30-31 and Lines 33-35. The sentences should be better linked.…. “

Authors’ response: All conceptions described in lines 33-39 have been checked in terms of content corrected and corrected according to Reviewer suggestion.

Comment 3: “…Suggestion: the small sentences “Yellow and orange colours are created by carotenoids and flavonoids” and “Purple, violet and blue colours are from anthocyanins” can be better written in one sentence “ Depending on the colour of the flowers, different phytochemical profiles were observed, with yellow and orange flowers displaying higher concentrations of carotenoids and flavonoids, while purple, violet and blue coloured flowers displaying higher quantities of anthocyanins.”….”

Authors’ response: Last Reviewer suggestion to replace two small sentences by one longer have been done in manuscript text (lines: 33-39).